# GSVD for Geometry-Grounded Dataset Comparison: An Alignment Angle Is All You Need

**Arthur Sobrinho**
Institute of Computing, UFRJ
arthursfr@ic.ufrj.br

**Eduarda Marques**
Institute of Computing, UFRJ
eduardasm@ic.ufrj.br

**João Paixão**
Institute of Computing, UFRJ
jpaixao@ic.ufrj.br

**Daniel S. Menasche**
Institute of Computing, UFRJ
sadoc@ic.ufrj.br

**Heudson Mirandola**
Institute of Mathematics, UFRJ
mirandola@im.ufrj.br

## Abstract

Geometry-grounded learning asks models to respect structure in the problem domain rather than treating observations as arbitrary vectors. Motivated by this view, we revisit a classical but underused primitive for comparing datasets: *linear relations* between two data matrices, expressed via the co-span constraint $Ax = By = z$ in a shared ambient space.

To operationalize this comparison, we use the generalized singular value decomposition (GSVD) as a joint coordinate system for two subspaces. In particular, we exploit the GSVD form $A = HCU$, $B = HSV$ with $C^\top C + S^\top S = I$, which separates shared versus dataset-specific directions through the diagonal structure of $(C, S)$. From these factors we derive an interpretable *angle score* $\theta(z) \in [0, \pi/2]$ for a sample $z$, quantifying whether $z$ is explained relatively more by $A$, more by $B$, or comparably by both.

The primary role of $\theta(z)$ is as a *per-sample geometric diagnostic*. We illustrate the behavior of the score on MNIST through angle distributions and representative GSVD directions. A binary classifier derived from $\theta(z)$ is presented as an illustrative application of the score as an interpretable diagnostic tool.

## 1 Introduction

Comparing datasets is a recurring problem across machine learning and data analysis. It arises when assessing dataset shift between training and deployment Sugiyama et al. (2007), when comparing representations learned by different models (Huang et al., 2007), and when diagnosing similarities and differences between classes or domains (Morcos et al., 2018). While modern practice often compares datasets indirectly (via trained models or embedding distances), such procedures can obscure *why* two datasets appear similar or different.

We take a complementary view: compare datasets through their *geometry*. Many real-world datasets concentrate near low-dimensional structures, exhibit partially shared latent factors, and contain directions that are specific to one domain or class. When datasets live in a common ambient feature space (raw pixels, sensor measurements, or fixed embeddings), their geometric relationship becomes meaningful in its own right.

**A relation primitive.** Let $A = [a_1, \ldots, a_p] \in \mathbb{R}^{d \times p}$ and $B = [b_1, \ldots, b_q] \in \mathbb{R}^{d \times q}$ denote two dataset matrices (columns are observations in $\mathbb{R}^d$). We characterize similarity through the linear relation

$$Ax = By = z, \tag{1}$$

where $x \in \mathbb{R}^p$ and $y \in \mathbb{R}^q$ are coefficients and $z \in \mathbb{R}^d$ is a shared ambient vector. We treat (1) as a primitive object: it encodes compatibility in the ambient space without requiring pointwise correspondences between samples or an invertible mapping between domains.

**GSVD yields a joint frame.** A classical tool for paired subspace analysis is GSVD (Edelman & Wang, 2020). Given $A$ and $B$ with the same row dimension $d$, one common GSVD form provides

$$A = HCU, \qquad B = HSV, \qquad C^\top C + S^\top S = I, \tag{2}$$

where $H \in \mathbb{R}^{d \times d}$ is invertible (or left-invertible in reduced variants), $U \in \mathbb{R}^{p \times p}$ and $V \in \mathbb{R}^{q \times q}$ are orthogonal, and $C$ and $S$ are diagonal or block-diagonal with nonnegative entries. Intuitively, $H$ defines a shared ambient reference frame, while $(C, S)$ encode how strongly each shared direction contributes to $A$ versus to $B$.

**A single alignment angle.** Our goal is to summarize the relative explanatory power of $A$ versus $B$ for a sample $z$ by a minimal, interpretable quantity. We define an *alignment angle* $\theta(z) \in [0, \pi/2]$ derived from the GSVD frame: $\theta(z) \approx 0$ means "more $A$", $\theta(z) \approx \pi/2$ means "more $B$", and $\theta(z) \approx \pi/4$ indicates shared structure.

**Contributions.** We summarize our contributions as follows:

1. We propose linear relations in the co-span form $Ax = By = z$ as a minimal, geometry-grounded primitive for dataset comparison.

2. We use GSVD as a natural joint coordinate system for comparing two subspaces, making shared vs. dataset-specific directions explicit via $(C, S)$.

3. We derive an interpretable angle score $\theta(z)$ that quantifies a per-sample diagnostic of relative dataset alignment and supports a binary classification.

4. We illustrate the geometric behavior of $\theta(z)$ on MNIST through angle distributions and GSVD-derived representative directions.

**Organization.** The remainder of the paper is organized as follows. Section 2 reviews related work, and Section 3 introduces the background on linear relations and GSVD. Section 4 defines the relative alignment angle $\theta(z)$, explains how to compute it efficiently in the GSVD frame, and describes downstream uses such as classification and diagnostics. Section 5 reports experiments on MNIST, including class-conditional angle distributions and representative GSVD directions. Section 6 discusses additional geometric interpretations, followed by conclusions in Section 7. The appendices providing additional derivations and supporting GSVD properties, including the span-view interpretation and proofs of the main technical results can be found in (de Souza Marques et al., 2026).

## 2 RELATED WORK

Our work sits at the intersection of (i) classical subspace comparisons and multivariate statistics, (ii) instance alignment objectives, and (iii) modern representation-similarity and transfer-learning perspectives.

**Subspace geometry and joint decompositions.** SVD provides canonical coordinates for a single matrix and underpins least-squares and low-rank approximation (Golub & Van Loan, 2013; Stewart, 1993). To compare two subspaces, principal angles quantify overlap via the singular values of $Q_A^\top Q_B$ for orthonormal bases $Q_A, Q_B$ (Björck & Golub, 1973; Knyazev & Argentati, 2002), and Canonical Correlation Analysis (CCA) identifies maximally correlated projections between variable sets (Hotelling, 1936). GSVD generalizes SVD to matrix pairs (Van Loan, 1976; Paige & Saunders, 1981), producing a joint frame $A = HCU$, $B = HSV$ where the diagonal factors $(C, S)$ expose shared versus dataset-specific directions and encode principal-angle-like structure through $(c_i, s_i)$. We use this joint frame as an interpretable comparative coordinate system and derive from it a per-sample alignment angle $\theta(z)$. Whereas principal angles describe geometry between subspaces, $\theta(z)$ provides a complementary *sample-level diagnostic* indicating whether an observation aligns more strongly with $A$, $B$, or the shared structure.

**Instance alignment and Procrustes.** When sample correspondences are available (or assumed), alignment is often formulated as minimizing an instance-wise discrepancy under a restricted transformation family. Procrustes analysis is the canonical example, aligning two point clouds by an

orthogonal (or similarity) transform (Gower, 1975). In many dataset-comparison settings, however, correspondences are ambiguous or unreliable, motivating our emphasis on relations and subspace geometry. Still, instance alignment is complementary to our approach: the GSVD angles provide a principled way to downweight weakly shared directions before performing alignment (de Souza Marques et al., 2026) (see Appendix C).

**Representation similarity and dataset distances.** Recent work compares learned representations using invariances to re-parameterizations. SVCCA and PWCCA adapt CCA to compare neural representations through truncated subspaces and weighting schemes (Raghu et al., 2017; Morcos et al., 2018), while CKA provides a robust similarity index widely used for layerwise comparisons (Kornblith et al., 2019; Cortes et al., 2012). In parallel, distributional comparators such as MMD (Gretton et al., 2012) and optimal transport (Peyré & Cuturi, 2019) measure dataset discrepancy directly, and FID is a practical feature-statistics score in generative modeling (Heusel et al., 2017). These approaches often output a *single* similarity value (or a layerwise matrix) and prioritize invariance or distribution matching. Our goal is complementary: we extract an explicit shared frame (via GSVD), interpretable representative directions, and a *per-sample* diagnostic $\theta(z)$.

**Instance-based transfer learning.** A different but related viewpoint is transfer learning by *reweighting* or *selecting* source instances to better match a target domain. Surveyed in Pan & Yang (2010), this line includes boosting-style methods such as TrAdaBoost (Zheng et al., 2020) and covariate-shift / importance-weighting approaches (Sugiyama et al., 2007; Huang et al., 2007). Our angle score $\theta(z)$ plays a conceptually similar role as a *relative compatibility* signal: it can be used to flag samples that are more consistent with one dataset than the other, which is useful for auditing, filtering, or prioritizing instances when comparing or transferring between domains.

## 3 BACKGROUND: LINEAR RELATIONS AND GSVD

This section presents two equivalent views of linear relations: the *co-span* formulation, which underlies the rest of the paper, and the *span* formulation, developed further in (de Souza Marques et al., 2026). We also introduce GSVD as a joint coordinate framework for expressing relations between datasets, highlighting the block structure of $(C, S)$ that separates shared from dataset-specific directions.

### 3.1 LINEAR RELATIONS: CO-SPAN VS. SPAN

A linear relation between the two subspaces can be formalized as

$$\mathcal{R}_{\text{co-span}} = \{(x, y) \in \mathbb{R}^p \times \mathbb{R}^q : Ax = By\}. \tag{3}$$

Here, $x$ and $y$ denote coordinate vectors, and the constraint imposes that both sides yield the *same* ambient vector $z := Ax = By \in \mathbb{R}^d$. From a geometric perspective, this encodes the intersection (and approximate intersection) structure between $\text{col}(A)$ and $\text{col}(B)$, which denote the column spaces of $A$ and $B$, respectively; it can be naturally interpreted as an *alignment of dimensions* by enforcing representability in a shared ambient space.

Equivalently, one may introduce an explicit shared latent parameter $w$:

$$\mathcal{R}_{\text{span}} = \{(x, y) : \exists\, w \text{ s.t. } x = Fw, \ y = Gw\}, \tag{4}$$

for suitable linear maps $F, G$. This view is closer to *instance alignment*, but in many dataset-comparison settings correspondences are unreliable, motivating the relation-first view (3).

### 3.2 GSVD AS A JOINT GEOMETRY

To compare two subspaces simultaneously, GSVD produces a *joint coordinate system*. In one common form, given matrices $A$ and $B$, GSVD produces the decomposition given by (2). Intuitively, $H$ defines a shared ambient reference frame, while the diagonal factors $C^{d \times p}$ and $S^{d \times q}$ quantify how strongly each shared direction contributes to $A$ and $B$. In particular, the diagonal (or block-diagonal) factors $C$ and $S$ are ordered so that entries of $C$ decrease and entries of $S$ increase, making "more-$A$" versus "more-$B$" directions explicit.

**Block structure of $C$ and $S$.** In the GSVD form (2), the diagonal (or block-diagonal) factors $C$ and $S$ can be written by isolating a central diagonal block (the shared subspace) and padding with identity/zero blocks:

$$C \;=\; \begin{bmatrix} I_r & 0 & 0 \\ 0 & \widetilde{C} & 0 \\ 0 & 0 & 0_t \end{bmatrix}, \qquad S \;=\; \begin{bmatrix} 0_r & 0 & 0 \\ 0 & \widetilde{S} & 0 \\ 0 & 0 & I_t \end{bmatrix}, \tag{5}$$

where $I_r$ and $I_t$ are identity matrices, $0_r$ and $0_t$ are zero matrices, and $\widetilde{C}, \widetilde{S} \in \mathbb{R}^{k \times k}$ are diagonal with strictly positive entries. The sizes satisfy $r + k + t = d$.

The $C$ and $S$ elements are ordered as follows:

$$c_{11} \geq c_{22} \geq \cdots \geq c_{dd},$$
$$s_{11} \leq s_{22} \leq \cdots \leq s_{dd}.$$

**Shared vs. specific directions.** Directions where a diagonal entry of $C$ dominates correspond to geometry primarily explained by $A$; directions where $S$ dominates are primarily explained by $B$; comparable magnitudes indicate shared structure.

## 4 A RELATIVE ALIGNMENT ANGLE $\theta(z)$ AND ITS USES

This section defines the alignment angle associated with a fixed instance, presents an algorithm for its computation, characterizes the instances that attain its maximum and minimum values, and presents two downstream applications.

### 4.1 DEFINING THE ALIGNMENT ANGLE

We now turn the relation between two datasets $A$ and $B$ into a single, interpretable score. For a given $z \in \mathrm{col}(A) \cap \mathrm{col}(B)$, i.e., for which the co-span relation $Ax = By = z$ is feasible, define the fiber

$$\mathcal{R}_{\text{co-span}}(z) \;=\; \{(x, y) \in \mathbb{R}^p \times \mathbb{R}^q : Ax = By = z\}. \tag{6}$$

If $z$ does not lie in this intersection, the relation is undefined and the sample is considered *non-relational* with respect to $(A, B)$.

**Definition 1** (Alignment angle). The alignment angle of $z$ given matrices $A$ and $B$ is defined as

$$\theta(z) \;:=\; \arctan\!\left(\frac{\|x\|_2}{\|y\|_2}\right) \;\in\; \left[0, \tfrac{\pi}{2}\right], (x, y) \in \mathcal{R}_{\text{co-span}}(z), x \perp \mathrm{Ker}(A), y \perp \mathrm{Ker}(B). \tag{7}$$

**Interpretation.** $\theta(z) \approx 0$ means $z$ is explained by $A$ with smaller coefficient norm than by $B$ ("more $A$"); $\theta(z) \approx \pi/2$ means the symmetric situation ("more $B$"); $\theta(z) \approx \pi/4$ indicates comparable explanatory strength (shared structure).

The constraints $x \perp \mathrm{Ker}(A)$ and $y \perp \mathrm{Ker}(B)$ pick the minimum-$\ell_2$ norm coefficients among all pairs satisfying $Ax = By = z$. Hence $\|x\|_2$ and $\|y\|_2$ are canonical "costs" to represent the same $z$ using $A$ versus $B$, and the ratio $\|x\|_2/\|y\|_2$ compares these costs. Using $\arctan$ maps this positive ratio to a bounded, symmetric score in $[0, \pi/2]$, with equal costs $\|x\|_2 = \|y\|_2$ giving $\theta = \pi/4$.

### 4.2 COMPUTING $\theta(z)$ IN THE GSVD FRAME

Next, we express the alignment score (1) in terms of the GSVD factors, obtaining a closed-form expression that is easy to compute and geometrically interpretable: $H$ defines a shared ambient basis, while $(C, S)$ weight the relative contributions of $A$ and $B$ along each direction.

Let $(H, C, S, U, V)$ denote the GSVD factors in (2). For an arbitrary sample $z \in \mathbb{R}^d$, compute its coordinates in the shared frame by

$$c(z) \;:=\; \underset{c \in \mathbb{R}^d}{\arg\min}\; \|Hc - z\|_2^2 \qquad \Rightarrow \qquad c(z) = H^\dagger z. \tag{8}$$

Define the GSVD-weighted costs

$$a(z) := \left\| C^\dagger c(z) \right\|_2, \qquad b(z) := \left\| S^\dagger c(z) \right\|_2, \tag{9}$$

where $\dagger$ denotes the Moore–Penrose pseudoinverse (optionally truncated for numerical stability). Because $U$ and $V$ are orthogonal, these costs correspond to the norms of canonical coefficient choices in $\mathcal{R}_{\text{co-span}}(z)$ (up to the same orthogonal transforms).

**Theorem 1** (GSVD produces alignment angles).

$$\theta(z) = \arctan\left(\frac{a(z)}{b(z)}\right) = \arctan\left(\frac{\left\| C^\dagger c(z) \right\|_2}{\left\| S^\dagger c(z) \right\|_2}\right), \tag{10}$$

where $c(z) = H^\dagger z$ and $(H, C, S, U, V)$ are the GSVD factors in (2).

*Proof.* See (de Souza Marques et al., 2026). □

## 4.3 Finding Extreme Directions

So far, the pipeline is *forward*: given a sample $z$, we compute $\theta(z)$ as a relative-alignment diagnostic. We can also ask the inverse question: *which vectors $z$ are maximally A-like or maximally B-like under the GSVD geometry?* Equivalently, we seek

$$z_{\max} \in \arg\max_{z \in \mathcal{Z}} \theta(z), \qquad z_{\min} \in \arg\min_{z \in \mathcal{Z}} \theta(z), \tag{11}$$

over a constraint set $\mathcal{Z}$ (e.g., $\|z\|_2 = 1$, or $z$ restricted to the span of training data). This viewpoint produces representative "extreme" directions for visualization and diagnostics, complementing the per-sample scoring setting of Section 4.2.

In what follows, we characterize $z_{\max}$ and $z_{\min}$ in terms of the GSVD structure. In particular, $z_{\max}$ and $z_{\min}$ correspond to specific columns of the shared matrix $H$: namely, $h_{r+k}$ and $h_{r+1}$ respectively. These vectors represent the last and first shared generalized components of the datasets, where "shared" in this instance refers to components associated with indices for which both diagonal matrices $C$ and $S$ have nonzero entries. As such the GSVD provides not only the coordinate system used to calculate (7), but that same basis encodes the solution for optimization problem (11).

Let $\tilde{x} = (\tilde{x}_1, \tilde{x}_2, \tilde{x}_3)$ and $\tilde{y} = (\tilde{y}_1, \tilde{y}_2, \tilde{y}_3)$, where the dimensions of the components of $\tilde{x}$ and $\tilde{y}$ correspond to the blocks of $C$ and $S$ given by (5), i.e., $\tilde{x}_1 \in \mathbb{R}^r$, $\tilde{x}_2 \in \mathbb{R}^k$ and $\tilde{x}_3 \in \mathbb{R}^t$. Similarly, $\tilde{y}_1 \in \mathbb{R}^r$, $\tilde{y}_2 \in \mathbb{R}^k$ and $\tilde{y}_3 \in \mathbb{R}^t$.

**Theorem 2** (GSVD produces extreme directions). Given datasets $A$ and $B$, the extreme directions $z_{\max}$ and $z_{\min}$ that maximize and minimize the alignment angle $\theta(z)$, respectively, are unique and given by $z_{\max} = Ax_{\max} = By_{\max}$ and $z_{\min} = Ax_{\min} = By_{\min}$, where $z_{\max} = h_{r+k}$, $z_{\min} = h_{r+1}$ and

$$x_{\max} = U^T \begin{bmatrix} 0 \\ \widetilde{S}\widetilde{C}^{-1}e_k \\ 0 \end{bmatrix}, \quad y_{\max} = V^T \begin{bmatrix} 0 \\ e_k \\ 0 \end{bmatrix}, \quad x_{\min} = U^T \begin{bmatrix} 0 \\ \widetilde{C}\widetilde{S}^{-1}e_1 \\ 0 \end{bmatrix}, \quad y_{\min} = V^T \begin{bmatrix} 0 \\ e_1 \\ 0 \end{bmatrix}.$$

$(H, C, S, U, V)$ are the GSVD factors in (2), $e_i$ is the $i$-th canonical basis vector, i.e., $(e_i)_j = 1$ if $j = i$ and $(e_i)_j = 0$ otherwise, and $h_i$ is the $i$-th column of matrix $H$.

*Proof.* See (de Souza Marques et al., 2026). □

**Remark 1** (Deflation / subsequent directions). A standard way to obtain subsequent maximizers is to augment the optimization problem underlying Theorem 2 with the additional constraint that $x$ be orthogonal to all previously computed maximizers. Let $X_{\text{prev}}$ denote the matrix whose columns are the previously obtained maximizers, and impose the constraint $X_{\text{prev}}^\top x = 0$ in the maximization problem. This procedure is commonly referred to as *deflation*, in analogy with extracting successive principal components in PCA. An explicit orthogonality constraint on $y$ is unnecessary, since its orthogonality is induced by that of $x$. For further details, we refer the reader to (Chu et al., 1997).

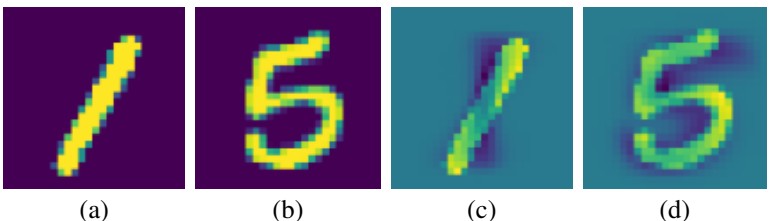

(a)        (b)        (c)        (d)

Figure 1: Representative MNIST digit samples for the GSVD pipeline. (a,b) raw digits "1" and "5"; (c,d) corresponding vectors after mean-centering.

## 4.4 DOWNSTREAM USES: DIAGNOSTICS AND ILLUSTRATIVE CLASSIFICATION

**Binary classification (relative geometry).** Given two domains (e.g., digit "1" vs. digit "5"), we build matrices $A$ and $B$ from their training samples (or low-rank bases thereof). For a test sample $z$, compute $\theta(z)$ and predict

$$\hat{\ell}(z) = \begin{cases} A, & \theta(z) \leq \tau, \\ B, & \theta(z) > \tau, \end{cases}$$

with $\tau = \pi/4$ as a symmetry default, or tuned on validation data. The pseudo-code of the classifier is presented in (de Souza Marques et al., 2026) (see Algorithm 1).

We emphasize that this rule is presented only as an illustrative consequence of the alignment score. The goal of the method is not to propose a competitive classifier, but to provide a geometrically interpretable diagnostic of how strongly a sample aligns with each dataset.

**Diagnostics.** Samples whose angles conflict with their nominal labels (e.g., a class-$A$ sample with $\theta(z)$ close to $\pi/2$) are candidates for auditing. The GSVD frame also provides interpretable directions, which can be used to visualize shared versus dataset-specific structure.

**Decomposition Complexity.** GSVD requires $O(d^3)$ time, which presents a bottleneck for large-scale databases. Since the decomposition is performed as a preprocessing step, however, it does not impact cost at inference time. Broader adoption and use cases of the GSVD in machine learning contexts may motivate the development of more efficient algorithms, leaving reduced decomposition complexity as a promising direction for future work.

**Numerical considerations.** In practical implementations the diagonal entries of $C$ and $S$ may contain very small values, making the pseudoinverses $C^\dagger$ and $S^\dagger$ sensitive to noise. In our experiments we use a truncated pseudoinverse that ignores singular values below a small tolerance. A systematic analysis of stability with respect to rank truncation, preprocessing, and noise remains an interesting direction for future work.

## 5 EXPERIMENTS ON MNIST

We describe a simple protocol on MNIST (LeCun et al., 1998) to illustrate how $\theta$ behaves as an angle score for classification. Additional results on MNIST-Fashion are reported in (de Souza Marques et al., 2026)(see Appendix E). The purpose of this experiment is not to benchmark classification performance but to illustrate the geometric behavior of the alignment score. MNIST provides a controlled setting in which the induced subspace structure can be visualized and interpreted. The same pipeline can be applied to other representations, including learned feature embeddings, which we leave for future work.

**Setup.** We choose two digits $a$ and $b$ (e.g., "1" vs. "5"). Each image is vectorized in $\mathbb{R}^{784}$ (from $28 \times 28$ pixels) and mean-centered. MNIST consists of 60,000 training images and 10,000 test images. We construct the matrix $A$ by stacking $p = 900$ training images from digit $a$ as columns, and analogously construct $B$ by stacking $q = 800$ training images from digit $b$, with the columns selected uniformly at random from the training set.

**Evaluation.** The histograms reported in Figure 2 are obtained by computing $\theta(z)$ for all test samples $z$ belonging to the two selected digits, i.e., for all available instances of digits $a$ and $b$ in the MNIST

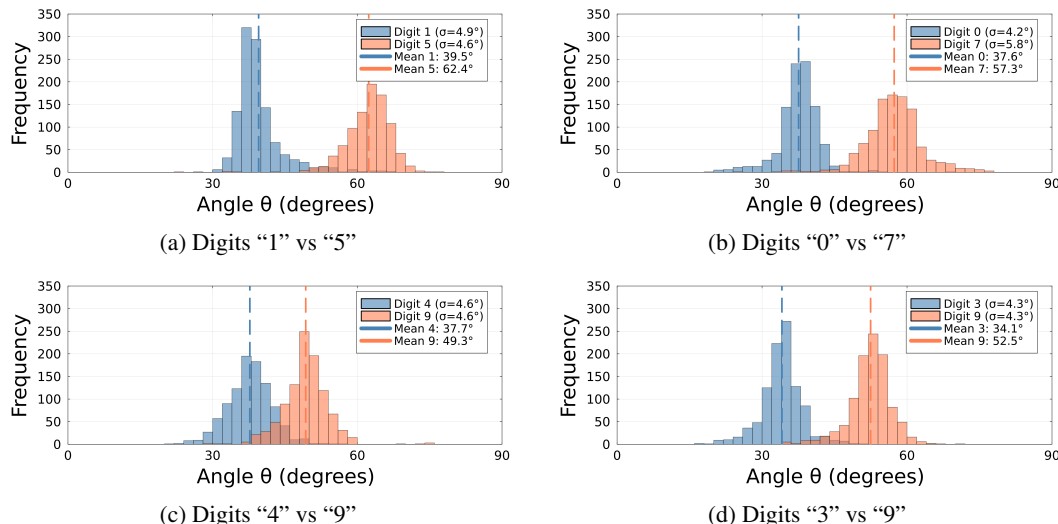

(a) Digits "1" vs "5"  (b) Digits "0" vs "7"

(c) Digits "4" vs "9"  (d) Digits "3" vs "9"

Figure 2: Empirical distributions of the angle $\theta(z)$ on the MNIST test set for four digit pairs. Values closer to $0$ indicate stronger alignment with the first digit in the pair, while values closer to $90°$ indicate stronger alignment with the second.

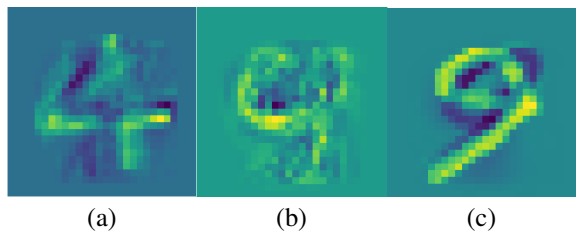

(a)            (b)            (c)

Figure 3: Representative $H$ component directions reconstructed in the image space as 28x28 images, using the viridis colormap, obtained by the GSVD-based optimization in Section 4.3: (a) representation of a solution from the minimization problem indicating a more "4-like" direction; (c) representation of the solution from the maximization problem indicating a more "9-like" direction; (b) reconstruction of a shared direction that encapsulates the structure of both "4" and "9".

test set, using **??**. We report the empirical class-conditional distributions of $\theta(z)$ as a diagnostic of relative geometric separability.

**Interpreting the $\theta(z)$ histograms.** If the two digits induce clearly different subspaces, the histograms concentrate on different angle ranges (small angles for the first digit in the pair and large angles for the second). Overlap between the histograms indicates directions that are similarly explained by both subspaces, and thus a more ambiguous relative geometry.

For a scalar, threshold-free summary of the separation between the class-conditional $\theta$ distributions, we report Fisher–Rao distances between the corresponding histograms in (de Souza Marques et al., 2026) (Table 2).

**Interpreting the $\theta(z)$ extremes.** The extreme columns correspond to directions of the shared GSVD frame $H$ that maximize or minimize the alignment angle $\theta(z)$ as derived in Section 4.3. The columns of $H$ can represent directions ordered by increasing relative-alignment angle $\theta(z)$, ranging from directions that are maximally aligned with $A$ to those that are maximally aligned with $B$. Intermediate columns capture directions that jointly reflect structural characteristics of both subspaces.

Figure 3 illustrates this behavior for matrices $A$ and $B$ constructed from MNIST images of digits 4 and 9, respectively. The $A$-aligned extreme exhibits a prototypical "4-like" structure with sharp, edge patterns, while the $B$-aligned extreme emphasizes rounded contours and a diagonal component. The

central image reveals a blended representation, combining elements shared by both digits, and thus illustrates directions along which samples from the two classes are similarly explained.

These visual structures are inherently data-dependent: the appearance of extreme and intermediate directions is determined by the specific instances used to construct $A$ and $B$. Consequently, they should be interpreted as empirical manifestations of the underlying class geometry rather than canonical templates.

## 6    DISCUSSION: GEOMETRY, SCALE, AND INFORMATION GEOMETRY

This section connects the alignment angle $\theta$ to probabilistic and information-geometric interpretations. While $\theta(z)$ was originally motivated by linear-algebraic considerations, it naturally induces probabilistic structures whose geometry can be studied using tools from information geometry.

**From angles to probabilistic geometry.** Our goal is to connect the alignment angle $\theta$, originally motivated by linear-algebraic arguments, to probabilistic and information-geometric interpretations. We do so in two complementary steps.

First, we show that each sample-wise angle $\theta(z)$ induces a Bernoulli posterior whose Fisher–Rao geometry is linear in $|\theta(z) - \theta(z')|$. Second, we move from individual samples to distributions over angles and study the Fisher–Rao distance between $\theta$-histograms, showing that this distance is driven primarily by mass near $\theta = \pi/4$, while samples near 0 or $\pi/2$ correspond to confident regions with negligible contribution.

The results in this section are presented using Fisher–Rao geometry. Alternatively, they can also be derived via the Hellinger geometry, as discussed in (de Souza Marques et al., 2026) (see Appendix G).

### 6.1    PER-SAMPLE FISHER–RAO GEOMETRY INDUCED BY $\theta$

$\theta(z) = \arctan(a(z)/b(z))$. We interpret inverse-squared costs as evidences and adopt the parametric posterior model

$$P(A \mid \theta) = \frac{1/a(z)^2}{1/a(z)^2 + 1/b(z)^2} = \frac{b(z)^2}{a(z)^2 + b(z)^2} = \cos^2 \theta, \qquad P(B \mid \theta) = \sin^2 \theta, \qquad (12)$$

so that $P(A \mid \theta)/P(B \mid \theta) = \cot^2 \theta$. This posterior coincides with a Bayes posterior based on likelihoods $p(\theta \mid A)$ and $p(\theta \mid B)$ whenever

$$\frac{p(\theta \mid A)}{p(\theta \mid B)} = \frac{\pi_B}{\pi_A} \cot^2 \theta, \qquad (13)$$

where $\pi_A = P(A)$ and $\pi_B = P(B)$.

Under the square-root (Bhattacharyya) embedding $\psi(\theta) = (\cos \theta, \sin \theta)$ of the Bernoulli simplex, the Fisher–Rao distance between the posteriors induced by two samples $z$ and $z'$ is

$$d_{\mathrm{FR}}(p(z), p(z')) = 2 \operatorname{acos}\big(\langle \psi(\theta(z)), \psi(\theta(z')) \rangle\big) = 2 \operatorname{acos}\big(\cos(\theta(z) - \theta(z'))\big) = 2|\theta(z) - \theta(z')|, \tag{14}$$

since $\theta \in [0, \pi/2]$. Thus, at the sample level, differences in $\theta$ correspond exactly to Fisher–Rao distances between induced Bernoulli posteriors.

### 6.2    FISHER–RAO DISTANCE BETWEEN $\theta$-HISTOGRAMS AND POSTERIOR AMBIGUITY

We now move from individual samples to distributions over angles. Fix a binning of $[0, \pi/2]$ into $m$ bins and let $P = (P_i)$ and $Q = (Q_i)$ be normalized histograms estimating the class-conditional likelihoods $P(\theta \in \text{bin } i \mid A)$ and $P(\theta \in \text{bin } i \mid B)$. Let $\pi_A$ and $\pi_B$ denote the class priors and define the mixture weights $m_i = \pi_A P_i + \pi_B Q_i$. For bin index $I = i$, define the binary label random variable $Y = \mathbf{1}\{\text{class} = A\}$, which equals 1 if the sample belongs to class $A$ and 0 otherwise. The posterior distribution of $Y$ given $I = i$ is

$$(Y \mid I = i) \sim \text{Bernoulli}(r_i), \qquad r_i = P(A \mid I = i) = \frac{\pi_A P_i}{\pi_A P_i + \pi_B Q_i}. \tag{15}$$

The Fisher–Rao distance between the histograms, denoted by $d_{\mathrm{FR}}(P,Q)$, and defined as $d_{\mathrm{FR}}(P,Q) = 2\arccos\left(\sum_{i=1}^{m}\sqrt{P_i Q_i}\right)$, satisfies

$$d_{\mathrm{FR}}(P,Q) = 2\arccos\left(\sum_{i=1}^{m} m_i \sqrt{\frac{r_i(1-r_i)}{\pi_A \pi_B}}\right) = 2\arccos\left(\frac{\mathbb{E}_{I\sim m}[\mathrm{Std}(Y|I)]}{\sqrt{\pi_A \pi_B}}\right). \tag{16}$$

Hence, $d_{\mathrm{FR}}(P,Q)$ is a monotone *decreasing* function of the *expected posterior standard deviation* $\mathbb{E}[\mathrm{Std}(Y\mid I)]$: bins with $r_i \approx \frac{1}{2}$ contribute most to the standard deviation, while bins with $r_i \approx 0$ or 1 contribute little. Under the parametric posterior (12) and equal priors, samples with $\theta \approx \pi/4$ concentrate probability mass in posterior bins with $r_i \approx \frac{1}{2}$, whereas samples with $\theta \approx 0$ or $\pi/2$ concentrate probability mass in bins with $r_i$ close to 1 or 0, respectively. This establishes a direct link between the linear-algebraic alignment angle $\theta$, the distance between class-conditional likelihoods, and posterior uncertainty.

**Numerical illustration.** Figure 2 provides an empirical illustration of the theoretical picture developed above. For digit pairs with clearer geometric separation (e.g., "1" vs. "5", "3" vs. "9" and "0" vs. "7"), the class-conditional $\theta$-histograms concentrate toward opposite extremes, with relatively little mass near $\theta = \pi/4$, indicating confident posterior regions. In contrast, visually similar pairs (e.g., "4" vs. "9") exhibit increased mass around $\theta \approx \pi/4$, reflecting higher posterior ambiguity. This behavior is consistent with the Fisher–Rao analysis: overlap near $\pi/4$ corresponds to bins with $r_i \approx \frac{1}{2}$ that dominate the Bhattacharyya coefficient and reduce histogram-level Fisher–Rao distance.

Together, the results presented in this section complete the connection between $\theta$ as introduced in Definition 1 and probabilistic notions of posterior uncertainty through Fisher–Rao geometry.

## 7    CONCLUSION

We introduced a geometry-grounded approach to dataset comparison built around a single primitive, the co-span relation $Ax = By = z$, and operationalized it through a GSVD joint frame. The resulting alignment angle $\theta(z)$ provides a minimal, interpretable summary of *relative* explanatory power: small angles indicate that $z$ is represented more economically by $A$, large angles favor $B$, and values near $\pi/4$ correspond to shared structure. Beyond yielding a simple illustrative decision rule, the same GSVD factors expose representative directions (including extremes) that act as visual diagnostics of what is shared and what is dataset-specific. Empirically, MNIST digit pairs exhibit characteristic class-conditional $\theta$ distributions whose overlap (or lack thereof) matches intuitive notions of separability. The Fisher–Rao distance between $\theta$-histograms complements these plots with a scalar, information-geometric separation metric, linking "dataset similarity" to posterior ambiguity induced by the angle.

This work is limited to two domains and controlled pixel-level datasets. Several extensions are left for future work. First, the framework can be generalized from dataset pairs to multiple domains, for instance by aggregating pairwise angles into a simplex-valued score or using multiway GSVD-style constructions. Second, the robustness of the alignment angle has not yet been characterized; future work should study the sensitivity of $\theta$ to preprocessing, rank selection, and the regularization or truncation of $C^{\dagger}$ and $S^{\dagger}$, as well as its behavior under noise and partial mismatch when $z \notin \mathrm{col}(A)\cap\mathrm{col}(B)$. Finally, while our experiments focus on raw pixel representations for interpretability, the framework applies more broadly to arbitrary feature spaces. In particular, matrices $A$ and $B$ can be constructed from feature embeddings produced by pretrained models such as convolutional networks or transformers. In such representations the linear subspace assumption may be more appropriate, since modern embeddings are designed to linearize semantic structure. Exploring the behavior of the alignment angle in such representation spaces is therefore a natural direction for future work.

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
