# OpenReview forum: "GSVD for Geometry-Grounded Dataset Comparison: An Alignment Angle Is All You Need"
_ICLR.cc/2026/Workshop/GRaM — ICLR 2026 Workshop GRaM Poster_

### Official Review · Reviewer_EZXW · 2026-02-08
**Interesting Geometric Primitive for Dataset Comparison, but Limited by Linear Assumptions and Scalability Concerns**

**Rating:** 6
**Confidence:** 4

**Review:**

SummaryThe paper proposes a geometry-grounded method for comparing two datasets ($A$ and $B$) by analyzing their joint generalized singular value decomposition (GSVD). By modeling datasets as subspaces in a shared ambient space, the authors derive a sample-wise "alignment angle" $\theta(z)$ based on the ratio of GSVD weights. This angle quantifies whether a sample is better explained by dataset $A$, dataset $B$, or shared structures. The method is validated on MNIST and Fashion-MNIST to demonstrate class separability and to visualize "extreme" directions.Strengths (Originality & Relevance)Relevance to GRaM: The paper is a perfect fit for the workshop's theme. It revisits classical linear algebra (GSVD) to provide a geometric interpretation of dataset relationships, moving away from "black-box" metrics.Novelty of the Primitive: Using the GSVD-derived angle as a granular, sample-level diagnostic score is an interesting and original contribution. Unlike global metrics (like FID or CKA), this approach allows for inspecting individual sample alignment, which has high potential for data curation and outlier detection.Theoretical Clarity: The mathematical derivation connecting the co-span relation ($Ax=By=z$) to the GSVD factors is rigorous and theoretically elegant.Weaknesses (Engineering Implementation & Technical Soundness)While the math is sound, I have significant concerns regarding the engineering feasibility and the gap between the theoretical assumptions and modern application scenarios:The "Linearity" Gap: The method relies on the assumption that datasets are well-represented by linear subspaces (column spaces). While this holds for simple datasets like MNIST (raw pixels), modern high-dimensional data (e.g., natural images) lies on non-linear manifolds.Critique: Applying this directly to raw pixels of complex datasets would likely fail. The paper would be significantly stronger if the authors demonstrated this method on extracted feature embeddings (e.g., ResNet or ViT features), where the linear subspace assumption is more plausible.Computational Complexity & Scalability:Critique: The engineering reality of GSVD is a major bottleneck. The computational complexity is roughly $O(d^3)$ or dependent on the number of samples. For modern embeddings ($d=4096$) or large datasets ($N > 50k$), computing the full GSVD is computationally heavy compared to lightweight baselines like cosine similarity of centroids.Missing Analysis: The paper lacks a discussion on wall-clock time and memory consumption. The claim of a "lightweight classifier" holds only during inference; the "training" (decomposition) phase is expensive.Numerical Stability of the Inverse:Critique: The calculation of alignment relies on $c(z) = H^{\dagger}z$. In real-world engineering, the matrix $H$ from GSVD is often ill-conditioned. The paper mentions "truncation" briefly but does not provide a robustness analysis. Small perturbations in the singular value threshold could drastically shift $\theta(z)$ for samples near the $\pi/4$ (shared) boundary.Comments on Writing & Results VerificationResults on "Toy" Data: Validating only on MNIST and Fashion-MNIST in 2026 is a weakness, even for a workshop. While the histograms (Fig 2) look accurate, they essentially confirm that distinct digits (1 vs 5) are linearly separable, which is already well-known.Visualization: Figure 1 and Figure 4 (vectorized images) are uninformative noise patterns. Showing the mean-centered difference or mapping the vectors back to the image space (by adding the mean) would provide better visual intuition.Notation: The switch between "co-span" and "span" views is mathematically precise but dense. The paper is well-organized, but the reliance on extensive appendices for proofs makes the main text difficult to self-contain.Questions for AuthorsHave you tested the stability of $\theta(z)$ with respect to the rank truncation parameter in $H^{\dagger}$?How does this method compare in computational time against standard subspace-based baselines (e.g., PCA-based distances) on larger datasets?Why was the method not applied to latent embeddings of more complex datasets (e.g., CIFAR-10 or ImageNet), where the geometry-grounded hypothesis would be more impactful?RecommendationThe paper presents a brilliant geometric idea but is limited by its "toy" experimental validation and linear assumptions. It is acceptable for the workshop as it stimulates discussion on geometric priors, but for the Proceedings track, the gap between theory and engineering practice (scalability/linearity) needs to be addressed.

**Pmlr Suitability:**

Yes

---

### Official Review · Reviewer_5ZCj · 2026-02-20
**Review of GSVD: An Alignment Angle is All You Need**

**Rating:** 6
**Confidence:** 4

**Review:**

Summary: The paper uses the generalized singular value decomposition (GSVD) to compare two datasets on a per-sample level. Through performing the GSVD on two datasets, they derive an angle score $\theta(z)$ quantifying whether a given sample $z$ is explained relatively more by $A$ or more by $B$. They demonstrate that $\theta(z)$ aligns with visual intuition on MNIST and that one can use it as a lightweight classifier.

Strengths:
- The paper is well-written and the derivation of $\theta(z)$ is quite clear. The GSVD is a natural tool for comparing datasets.
- It's useful that $\theta(z)$ can be used for per-sample comparison rather than solely at a global dataset level.

Weaknesses/Questions:
- It would be more useful to extend to non-pixel data, do the authors have an idea of how this could be done?
- The experiments on MNIST align with intuition but are limited, are there any other more complex image datasets that are underexplored? I think this could be useful in scientific datasets (e.g. cosmological data or turbulence are two domains that come to mind).
- I am somewhat confused about Figure 3/what it is showing. I am not sure this would be very useful in a more complex dataset.

Overall, I recommend acceptance as the paper provides a clean framework for geometrically comparing two datasets and I believe the work will be of interest to the GRAM community. I think it could benefit from applications to more complex datasets (e.g. scientific datasets that can be represented in a pixel format).

**Pmlr Suitability:**

Yes

---

### Official Review · Reviewer_8mFD · 2026-02-22
**Review of GSVD for Geometry-Grounded Dataset Comparison: An Alignment Angle Is All You Need**

**Rating:** 7
**Confidence:** 4

**Review:**

This paper introduces a novel, geometry-grounded framework for comparing two datasets. It proposes the co-span linear relation, \(Ax = By = z\), as a fundamental and interpretable way to link two datasets \(A\) and \(B\) through a shared vector \(z\) in the ambient space. This moves away from methods that require explicit instance correspondences.
It leverages the Generalized Singular Value Decomposition (GSVD) to create a joint coordinate system (\(H\)) for the two datasets. The diagonal factors (\(C, S\)) from the GSVD explicitly separate directions in the data that are shared versus those that are specific to one dataset.
It derives a per-sample angle score \(\theta(z) \in [0, \pi/2]\). This score quantifies the relative alignment of a new sample \(z\) to the two datasets: values near 0 mean it is more like dataset \(A\), near \(\pi/2\) more like dataset \(B\), and near \(\pi/4\) indicate shared structure. This provides a simple, scalar diagnostic for dataset membership.
The paper demonstrates how \(\theta(z)\) can be used for binary classification with a simple threshold. It also shows how the GSVD framework can be used to find "extreme" directions (\(z_{min}, z_{max}\)), which can be visualized as prototypical examples of what is unique to each dataset and what is shared between them.
The paper establishes a theoretical link between the linear-algebraic angle \(\theta\) and probability theory, showing that differences in \(\theta\) map directly to the Fisher-Rao distance between the induced Bernoulli posteriors. This provides a deeper interpretation of \(\theta\) in terms of posterior uncertainty and class separability.

The improvement suggested for this paper includes:

1. The connection to principal angles and Canonical Correlation Analysis (CCA) (Björck & Golub, 1973; Hotelling, 1936) is not sufficiently differentiated. The principal angles between subspaces are fundamentally defined by the singular values of \( (Q_A^\top Q_B) \), where \(Q_A\) and \(Q_B\) are orthonormal bases for the column spaces of \(A\) and \(B\). The GSVD factors \((C, S)\) are intimately related to these angles (specifically, \(\cos(\theta_i) = c_i / \sqrt{c_i^2 + s_i^2}\) and \(\sin(\theta_i) = s_i / \sqrt{c_i^2 + s_i^2}\)). The paper's "alignment angle" \(\theta(z)\) for a *sample* is a weighted combination of these subspace angles, but the core idea of a "shared frame" is already present in CCA. The paper would be strengthened by explicitly deriving the relationship between its \(\theta(z)\) and the canonical correlations and variates. As it stands, the GSVD frame may be presented as more novel than it is, given the long history of CCA and principal angles in this exact context.

2.  The paper argues its goal is "complementary" to methods that output a single similarity value like CKA (Kornblith et al., 2019). While complementary, the paper does not provide a strong empirical or theoretical argument for *why* its per-sample diagnostic is preferable to the insights one can get from CKA or SVCCA. CKA can be used to compare representations at different layers, providing a rich picture of similarity. The paper's approach, as presented, only compares the subspaces defined by the entire datasets (e.g., all "1"s vs. all "5"s). It doesn't show how to compare different *representations* of the same data, which is a major use case for CKA. The paper's contribution in this space is valid but niche, and the lack of a comparative experiment with CKA or SVCCA on a representation-similarity task is a notable omission.

3. In methodology, the "classification" use-case is presented only as an illustration, not as a proper experiment. There are no reported classification accuracies, no comparison to baseline methods (e.g., a simple k-NN classifier in pixel space, a linear SVM, or a classifier based on CCA features), and no error bars. The claim that \(\theta(z)\) supports "lightweight classification" is unsubstantiated without such a comparison. The "diagnostics" are purely visual. The paper does not quantify, for example, how the "outlier" in Figure 6(c) would be detected or what the practical impact of such diagnostics might be.

4.  The paper uses matrices \(A \in \mathbb{R}^{d\times p}\) and \(B \in \mathbb{R}^{d\times q}\) directly from 900 and 800 samples. For high-dimensional data where \(p, q < d\) (which is the case for MNIST with 784 dimensions), the column spaces are at most rank \(p\) and \(q\). The pseudoinverses \(C^\dagger\) and \(S^\dagger\) are used. The paper does not discuss the crucial issues of numerical stability, the effective rank of these matrices, or any regularization applied when inverting or taking pseudoinverses of near-singular matrices. This is a critical gap for the methodology's practical application. The method relies on projecting a test sample \(z\) onto the shared GSVD frame via \(c = H^\dagger z\). If \(z\) is not in the union of the column spaces of \(A\) and \(B\), this projection is a least-squares approximation. The paper does not analyze the effect of this approximation error on the resulting \(\theta(z)\). How robust is \(\theta(z)\) to noise or components of \(z\) that are orthogonal to both subspaces?

5.  The numerical analysis consists entirely of plotting histograms. There is no attempt to quantify the separation (beyond the one Fisher-Rao distance in Table 2, which is presented without context or comparison). The "extreme directions" visualizations in Figure 3 are interesting but purely anecdotal. There is no attempt to quantify what "A-like" or "B-like" means, or to compare these extreme images to, say, the mean image of each class.

**Pmlr Suitability:**

No

---

### Meta-Review · Area_Chair_i12W · 2026-02-25

**Decision:**

Accept

**Metareview:**

The paper is of interest to the GRaM community, and presents an elegant mathematical method for comparing two datasets. The key weakness is the omission of more complicated datasets than MNIST, which were not tested (and where reviewers predict that the method may fail). The reviewers make several good suggestions to improve the methodology, such as comparing to baseline classification methods, as well as an area for improvement in presentation: a more clear comparison to CCA. Nonetheless, the paper appears to be technically sound, interesting, and novel. Each reviewer recommended to accept and that the paper is worth presenting to the workshop community.

**Relevance To Proceedings:**

Yes — suitable for PMLR (long paper)

**Relevance To Workshop:**

Yes — suitable for GRaM

---

### Decision · Program_Chairs · 2026-03-02

Accept (Poster)